# Effect of Poly-l-Lysine Polycation on the Glucose Oxidase/Ferricyanide Composite-Based Second-Generation Blood Glucose Sensors

**DOI:** 10.3390/s19061448

**Published:** 2019-03-25

**Authors:** Ming-Jie Lin, Ching-Chou Wu, Ko-Shing Chang

**Affiliations:** 1Department of Bio-industrial Mechatronics Engineering, National Chung Hsing University, No. 145, Xingda Rd., South Dist., Taichung City 402, Taiwan; y4ky5k@yahoo.com.tw; 2Innovation and Development Center of Sustainable Agriculture, National Chung Hsing University, No. 145, Xingda Rd., South Dist., Taichung City 402, Taiwan; 3Technology Development Division, Tyson Bioresearch Inc., 5F, No. 16, 18, 20, 22, Kedong 3rd Rd., Zhunan Township, Miaoli County 35053, Taiwan; roger.chang@tysonbio.com

**Keywords:** second generation, glucose biosensors, α-poly-l-lysine, ferricyanide, point-of-care testing, long-term stability

## Abstract

Second-generation glucose biosensors are presently the mainstream commercial solution for blood glucose measurement of diabetic patients. Screen-printed carbon electrodes (SPCEs) are the most-used substrate for glucose testing strips. This study adopted hydrophilic and positively charged α-poly-l-lysine (αPLL) as the entrapment matrix for the immobilization of negatively charged glucose oxidase (GOx) and ferricyanide (FIC) on SPCEs to construct a disposable second-generation glucose biosensor. The αPLL modification is shown to facilitate the redox kinetics of FIC and ferrocyanide on the SPCEs. The SPCEs coated with 0.5 mM GOx, 99.5 mM FIC, and 5 mM αPLL had better sensitivity for glucose detection due to the appreciable effect of protonated αPLL on the promotion of electron transfer between GOx and FIC. Moreover, the SPCEs coated with 0.5 mM GOx, 99.5 mM FIC, and 5 mM αPLL were packaged as blood glucose testing strips for the measurement of glucose-containing human serum samples. The glucose testing strips had good linearity from 2.8 mM to 27.5 mM and a detection limit of 2.3 mM. Moreover, the 5 mM αPLL-based glucose testing strips had good long-term stability to maintain GOx activity in aging tests at 50 °C.

## 1. Introduction

Diabetes mellitus is the most common endocrine disease and is a disorder of carbohydrate metabolism producing high glucose levels in the blood, leading to different dysfunctions such as nerve degeneration, kidney failure, and blindness [1,2,3]. Diabetic patients must take precise daily measurements of their blood glucose concentrations. Presently, several methods, including metamaterial-based electromagnetic spectroscopy [4], fluorescence [5], near-infrared spectroscopy [6], and electrochemistry [7], have been developed for glucose detection. Furthermore, most commercial glucose biosensors adapt electrochemical methods via the catalysis of glucose oxidase (GOx) or glucose dehydrogenase to specifically detect glucose concentration [8]. A variety of self-monitoring glucose biosensors have been commercialized for point-of-care testing (POCT) [9,10,11]. Particularly, second-generation glucose biosensors using mediators dominate the POCT product market due to their low cost and good sensing properties with resistance to the impact of dissolved oxygen [12].

Most second-generation blood glucose sensors use ferrocene derivatives [13], ferricyanide (FIC), [14,15,16] and hydroquinone (HQ) [17] as a mediator for GOx to receive the electron of glucose oxidation [18,19,20,21,22,23,24,25]. These mediators assume the role of oxygen molecules, oxidizing the reduced-form flavin adenine dinucleotide (FADH_2_) of GOx. The FIC mediator dissolves faster and more completely in water than HQ and ferrocene, resulting in increased sensitivity, a wider linear range, and shorter response time [17]. However, both FIC and GOx (isoelectric point of 4.2) are negatively charged in a physiological solution (pH 7.4), implying that the repulsive electrostatic force is adverse to the electron transfer between FIC and GOx [16]. Therefore, the FIC concentration must be increased to several hundreds of millimolars to improve the electron transfer rate between GOx and FIC [26].

To counterbalance the repulsive electrostatic force between FIC and GOx, different polycations (PCs), such as poly[bis(2-chloroethyl)ether-*alt*-1,3-bis [3-(dimethylamino)-propyl]urea] and poly[oxyethylene-(dimethylimino)propyl(dimethylimino)ethyl hydrogensulfate] of quaternary ammonium; ethoxylated polyethylenimine (PEI) of secondary ammonium; and *ε*-poly-l-lysine (*ε*PLL), α-poly-l-lysine (αPLL), poly(allylamine) [16], and chitosan (CS) of primary ammonium [27] have been used to increase the electron transfer rate between FIC and GOx. Moreover, positively charged polyammonium can increase the long-term stability of negatively charged enzymes such as bilirubin oxidase [28,29] and GOx [30]. Nevertheless, Uematsu et al. (2012) found that the PCs mentioned above, except for the *ε*PLL and αPLL, frequently produced sedimentation or colloidal particles due to the formation of insoluble polyion complexes after the addition of FIC [16]. This inhibits the uniform spread of the GOx/FIC/PCs composite on the surface of electrodes for the industrial production of biosensors. Uematsu et al. found that a glucose biosensor fabricated by covalently entrapping GOx in an *ε*PLL matrix via glutaraldehyde on a glassy carbon electrode had an improved catalytic response in a more acidic solution, such as the pH 5.0 acetate buffer [31]. Although Uematsu et al. proved that the sensitivity of the εPLL/GOx-modified sensor was larger than that of an αPLL/GOx-modified sensor, the current response measured at the εPLL/GOx-modified sensor was significantly affected by varied pH values [31]. Compared with *ε*PLL (pKa = 7.6) [16], αPLL (pKa = 10.3) possesses more protonation and higher solubility in a physiological solution (pH 7.4). Moreover, due to its good biocompatibility, high solubility, and easy protonation, αPLL has been used as an entrapping matrix for glucose biosensors [32,33]. Wang and Chen found that a glassy carbon electrode (GCE) coated with a mixture of αPLL and GOx and then Nafion can produce mediatorless direct electron transfer (DET) without the use of nanomaterials [32]. Vilian et al. used a MnO_2_-decorated chemically reduced graphene oxide film modified GCE as an electrode substrate and coated the GOx–αPLL mixture on the electrode to form a DET glucose sensor [33]. The results suggest that αPLL has good biocompatibility for the protection of GOx. However, few studies have examined the effect of αPLL on the sensing properties of GOx/FIC composite-based second-generation glucose biosensors.

In this study, αPLL was used as the entrapment matrix for the immobilization of negatively charged GOx and FIC on screen-printed carbon electrodes (SPCEs) to construct disposable glucose biosensors. SPCEs are the most widely used commercial glucose test strip and generally consist of graphite, carbon black, and a polymer binder, which are hydrophobic and negatively charged. The hydrophilic coating and positively charged αPLL could affect the electrochemical properties of the SPCEs. Therefore, the effect of the αPLL coating layer on the electrochemical behavior of FIC in a pH 7.0 solution at the electrode/electrolyte interface of αPLL-modified SPCEs was explored using cyclic voltammetry (CV) and electrochemical impedance spectroscopy (EIS). Moreover, the sensing properties of GOx/FIC/αPLL-based glucose biosensors were determined in detail.

## 2. Materials and Methods

### 2.1. Reagents

Glucose oxidase (EC 1.1.3.4, 149,700 U/g, from *Aspergillus niger*), sodium dihydrogenphosphate dihydrate, sodium phosphate dibasic anhydrous, potassium chloride (KCl), and αPLL (*M*w = 150,000–300,000) were purchased from Sigma-Aldrich. FIC, ferrocyanide (FOC), D(+)-glucose, and sodium hydroxide were obtained from Showa. All chemicals were of reagent grade, were used without further purification, and were prepared in pure water purified through a Milli-Q system. Phosphate buffer solution (PBS) was prepared through the mixture of Na_2_HPO_4_ and NaH_2_PO_4_, and the pH value of the PBS was adjusted by 3 M NaOH.

### 2.2. Preparation of GOx/FIC/αPLL-Modified SPCEs

SPCEs supplied by Tyson Bioresearch Inc. (Miaoli, Taiwan) were used as the working electrode with a sensitive area of 3.64 mm^2^. To increase the electrochemical activity of the electrodes, the SPCEs were activated by preoxidizing [34,35], as mentioned below. SPCEs were first electrochemically cleaned in 100 mM PBS (pH 7.0) from −0.2 V to 1.3 V at a scanning rate of 0.1 V/s for 10 cycles with a potentiostat (1000A, CH Instruments, Austin, TX, USA) in a three-electrode system. An Ag/AgCl electrode (3 M NaCl) (RE-1B, BAS, Tokyo, Japan) and a Pt wire were respectively used as the reference electrode and the counter electrode. Subsequently, the cleaned SPCEs were electrochemically oxidized at 2 V for 300 s in 100 mM NaOH to increase the edge-plane-to-basal-plane ratio of the graphite, facilitating the electron transfer rate [34,35]. Concentration-varied αPLL, GOx, and FIC were respectively prepared in 25 mM PBS (pH 7.0). A 6 μL aliquot of GOx/FIC/αPLL mixture was dripped on the preoxidized SPCEs and kept in an oven at 25 °C for 30 min to form the sensing layer of the glucose biosensors. Nafion (5%) was dripped on the GOx/FIC/αPLL-coated SPCEs, and kept in an oven at 25 °C for 30 min.

### 2.3. Electrochemical Measurement

The redox behavior of FIC was estimated by CV in the range from +0.6 V to −0.2 V. The electron transfer kinetics and diffusive behavior of FIC and FOC at the electrode/electrolyte interface were evaluated using an IM-6 impedance analyzer (Zahner Electrik GmbH, Kronach, Germany). EIS measurement was carried out in the frequency range of 0.1 Hz to 100 kHz at the +0.217 V potential added with a 5 mV amplitude sine wave versus the Ag/AgCl reference electrode. The IM-6/THALES software package was used to analyze the impedance spectra and the simulation of equivalent circuits. Chronoamperometry was used to estimate the calibration curve of the glucose biosensor by applying the potential of +0.4 V in 25 mM PBS (pH 7.0) containing 99.5 mM FIC and 100 mM KCl.

### 2.4. UV–Visible Spectroscopy

The ultraviolet–visible (UV–visible) spectrum was recorded using a microplate spectrophotometer (Epoch™, Biotek, Winooski, VT, USA). Respective aliquots of 5 µL of GOx, αPLL, FIC, GOx/αPLL mixture, and FIC/αPLL mixture was dripped onto a micro-volume plate to determine the absorption spectra.

### 2.5. Stability of GOx/FIC/αPLL-Modified SPCE Strips

A 2 μL mixture of 5 mM αPLL, 0.5 mM GOx, and 99.5 mM FIC was dripped onto a commercial two-electrode SPCE with a sensitive area of 1.12 mm^2^, supplied by Tyson Bioresearch Inc. After bonding a spacer and a hydrophilic cover, the packaged SPCEs were formed into disposable glucose biosensor strips. These biosensor strips were kept at 50 °C to perform aging tests prior to glucose measurement. Human whole blood was obtained from volunteers and centrifuged at 3000 rpm for 10 min to obtain serum. The 120 mg/dL (6.67 mM) glucose-containing serum, verified using a commercial glucose analyzer (YSI 2300 STAT, YSI Inc., Yellow Springs, OH, USA), was used to measure the long-term stability of the glucose biosensor strips after different aging periods. Moreover, the current response of each testing strip was measured by chronoamperometry at +0.25 V for 10 s.

## 3. Results and Discussion

### 3.1. Effect of αPLL on FIC

Prior to exploring the sensing properties of the GOx/FIC/αPLL-based glucose biosensors, the effect of the αPLL coating layer on the electrochemical properties of FIC should be clarified. Generally, the protonated αPLL can attract negatively charged FIC through electrostatic force. Aliquots of 6 μL of 0.1, 0.25, 0.5, 2.5, and 5 mM αPLL prepared in distilled water were dripped onto preoxidized SPCEs and dried at 25 °C for 30 min. Subsequently, the αPLL-modified SPCEs were measured in 10 mM FIC solution (pH 7.0 adjusted by 1 M NaOH or HCl) using CV, as shown in Figure 1a. All the cyclic voltammograms present the well-defined redox behavior of FIC and FOC. The FIC solution was prepared in distilled water (pH 7.0) to enhance the electrostatic interaction with the αPLL layer. Figure 1b shows the corresponding cathodic peak current (*I_pc_*) and potential (*E_pc_*) and the anodic peak current (*I_pa_*) and potential (*E_pa_*) as a function of the αPLL concentration. The *I_pc_* and *I_pa_* measured at the preoxidized SPCEs, defined as *I_pc-SPCE_* and *I_pa-SPCE_*, are smaller than those obtained at the αPLL-modified SPCEs. Moreover, the *I_pc-SPCE_* is almost identical to the *I_pa-SPCE_*, suggesting that the redox reaction of FIC produces Nernstian reversible behavior [36]. The *I_pc_* and *I_pa_* measured at the αPLL-modified SPCEs, respectively defined as *I_pc-PLL_* and *I_pa-PLL_*, increased obviously with the αPLL concentration, indicating that the αPLL modifying layer can effectively promote the FIC concentration at the electrode surface. This could be attributed to the increase of the positive surface charge density of αPLL-modified SPCEs to enrich FIC in the αPLL modifying layer. It is worth noting that the *I_pc-PLL_* and the *I_pa-PLL_* exhibit different increasing slopes with the αPLL concentration. The *I_pc-PLL_* measured at the 0.1 mM αPLL-modified SPCEs is significantly larger than *I_pc-SPCE_*, but *I_pc-PLL_* in the range of 0.1 mM to 5 mM αPLL only presents a slight increase. In contrast, *I_pa-PLL_* increases obviously with the 0.1–5 mM αPLL coating. This is attributed to the fact that tetravalent FOC has stronger stoichiometric interaction with the protonated αPLL layer than does trivalent FIC [29]. This results in the GOx/FIC/αPLL-based glucose test strips obtaining a larger signal when GOx catalyzes FIC to FOC with the addition of glucose. Simultaneously, the larger electrostatic force between αPLL and FOC reduces the interaction between FIC and αPLL. As a result, *I_pc-PLL_* did not significantly increase after the αPLL coating.

Furthermore, the *E_pc_*, as shown in Figure 1b, the *E_pa_*, and the peak separation (Δ*E_p_* = *E_pc_* − *E_pa_*) of the αPLL-coated SPCEs indicate the change in the electron transfer kinetics. The 0.5 mM αPLL-coated SPCEs had a smaller Δ*E_p_* (218 mV), a more negative *E_pa_*, and a more positive *E_pc_* than did the preoxidized SPCEs and the 5 mM αPLL-coated SPCEs. The results imply that the 0.5 mM αPLL coating layer can lower the overpotential of FIC and FOC [37]. In particular, the *E_pa_* obtained at the 5 mM αPLL-coated SPCEs had the largest overpotential, attributed to a chemical–electrochemical mechanism. The surface charge density of the 5 mM αPLL layer increases the interactive force with FOC, making FOC difficult to oxidize. Figure 2a–c respectively show the cyclic voltammograms measured with a different scanning rate (*υ*) at the preoxidized SPCEs and at the 0.5 mM and 5 mM αPLL-coated SPCEs. The insert shows *I_pc-PLL_* as a function of *υ* with good linearity, implying a diffusion-controlled mechanism [38]. Figure 2d shows the corresponding *E_pc_* and *E_pa_* versus the logarithm of *υ*. The electron transfer rates (*ks*) of FIC reduction obtained on the preoxidized SPCEs and on the 0.5 mM and 5 mM αPLL-coated SPCEs were, respectively, 0.24, 0.30, and 0.21 s^−1^, with respective transfer coefficients (*α*) of 0.37, 0.54, and 0.49, as calculated from Equations (1) and (2):(1)$$E_{pc}=E^{0^{\prime }}-\;\frac{2.3RT}{\alpha nF}\log\nu$$
(2)$$\log k_{s}=\alpha \log(1-\alpha )\;+\;(1-\alpha )\log\alpha -\log\frac{RT}{nF\nu \;}-\frac{(1\;-\;\alpha )\alpha nF\Delta E_{P}}{2.3RT}$$
where *R* is the gas constant, *T* is the absolute temperature, *n* is the number of electrons in the electron transfer, and *F* is the Faradaic constant. The 0.5 mM αPLL-coated SPCEs had a larger *ks*, suggesting that the 0.5 mM αPLL coating is more effective than the 5 mM αPLL coating in increasing the electron transfer rate of FIC.

Furthermore, EIS was used to estimate the kinetics and diffusive behavior. Figure 3 shows the Nyquist plots obtained for the preoxidized SPCEs and for the 0.5 mM and 5 mM αPLL-coated SPCEs. All Nyquist plots consist of a linear part at lower frequencies, related to the diffusion-controlled behavior of FIC/FOC, and a semicircular part at high frequency, related to the kinetic reaction of electron transfer from FIC/FOC to the electrode. Moreover, the radius of the semicircle, related to the electron transfer resistance (*R_et_*), decreased with the PLL modification. The decreasing impedance of the solution/electrode interface is attributed to the stronger stoichiometric affinity of FOC/FIC with the protonated αPLL than with preoxidized SPCEs. These Nyquist plots were fitted using IM-6/THALES software to obtain the element values via the Randles equivalent circuit to explain the EIS behavior [39]. The Randles equivalent circuit includes the ohmic resistance (*R_s_*) of the electrolyte solution, the Warburg impedance (*Z_w_*) resulting from the ionic diffusion from bulk electrolyte to the electrode interface, the constant phase element (*CPE*), and the *R_et_*. Table 1 compares the fitting results related to the equivalent circuit elements for the preoxidized SPCEs and for the 0.5 and 5 mM αPLL-modified SPECs. The result shows that *R_et_* decreased as the αPLL concentration increased, attributed to the fact that the αPLL modifying layer can effectively promote the FIC/FOC concentration of the electrode surface. This is consistent with the CV result in Figure 1. Furthermore, the 0.5 mM αPLL-modified SPCEs have a smaller *Z_w_* than the preoxidized SPCEs and the 5 mM αPLL-modified SPCEs, suggesting that the FIC/FOC in the 0.5 mM αPLL coating has better diffusive transport than do those in the 5 mM αPLL layer.

### 3.2. Effect of FIC Concentration

As mentioned above, the αPLL coating affects the electrochemical properties of FIC and FOC mainly due to electrostatic interaction. Theoretically, when the FIC concentration exceeds the positive charge density of the αPLL layer, the surface concentration of FIC should be mainly determined by chemical potential. Figure 4a shows the cyclic voltammograms measured on the preoxidized SPCEs and 0.5 mM αPLL-modified SPCEs in concentration-different FIC solutions. The ratio of *I_p-PLL_* to *I_p-SPCE_* is presented in Figure 4b.

The lower FIC solution makes the *I_pa-PLL_*/*I_pa-SPCE_* larger than the *I_pc-PLL_*/*I_pc-SPCE_*, suggesting the stronger interaction force between protonated αPLL and tetravalent FOC enriching FOC in the αPLL layer. Although the *I_p_* increased with the FIC concentration, the higher FIC concentration causes a smaller *I_p-PLL_*/*I_p-SPCE_*. This result suggests that the higher FIC concentration reduces the electrostatic influence of protonated αPLL on FIC and FOC. Measured in 40 mM FIC solution, the *I_p-PLL_*/*I_p-SPCE_* was 1, indicating that the 0.5 mM αPLL coating has little influence on the redox behavior of FIC/FOC. Therefore, the concentration ratio of FIC to αPLL must be below 80 to obtain better electrostatic influence.

### 3.3. Effect of Other Anions

Protonated αPLL has been shown to facilitate the electron transfer kinetics and diffusive behavior of FIC/FOC due to the electrostatic interaction. Katano et al. found that the anions of monovalent H_2_PO_4_^−1^ and divalent HPO_4_^−2^ could influence the equilibrium status of εPLL-FIC/FOC complex in a PBS solution to induce the sedimentation of the εPLL-FIC/FOC complex [29,37]. This study used anions with different ionic strength levels to estimate the effect on the reductive current of FIC. Figure 5 shows the *I_pc_* obtained in different 10 mM FIC-containing solutions. As found previously, the 0.5 mM αPLL-modified SPCEs can obtain an *I_pc_* in the 10 mM FIC, which is significantly larger than that for the preoxidized SPCEs in the 10 mM FIC. Although the addition of 100 mM KCl increased the solution conductivity to 14.39 mS/cm, the high concentration of monovalent Cl^−1^ did not affect the interaction of protonated αPLL and trivalent FIC due to the lower valency product of Cl^−1^. It is worth noting that the *I_pc_* obtained in the mixture of 10 mM FIC, 100 mM KCl, and 10 mM PBS (pH 7.0) is greater than that measured in the 10 mM FIC and 100 mM KCl solution due to the buffering capacity of PBS maintaining more protonation of the αPLL layer than the FIC/KCl solution. Furthermore, increasing the PBS concentration and the solution conductivity respectively to 25 mM and 19.20 mS/cm did not significantly change *I_pc_*, indicating that the buffering capacity of 10 mM PBS can protonate most amine groups of αPLL and that the H_2_PO_4_^−1^ and HPO_4_^−2^ did not influence the interaction of protonated αPLL and trivalent FIC. As a result, the mixture of 10 mM FIC, 100 mM KCl, and 25 mM PBS was used as the testing solution to estimate the enzyme properties of the αPLL-coated glucose biosensors.

### 3.4. Sensing Properties of GOx/FIC/αPLL-Modified SPCEs

The αPLL modification significantly increased the redox rate of FIC/FOC on SPCEs. It is potentially interesting to explore the effect of αPLL on the interaction between GOx and FIC. The different concentration ratios of GOx and FIC to αPLL were immobilized on preoxidized SPCEs to estimate the GOx activity. The sensing properties of GOx/FIC/αPLL-modified SPCEs were estimated at +0.4 V versus an Ag/AgCl reference electrode in 25 mM PBS (pH 7.0) containing 100 mM KCl, and the FIC concentration was the same as the FIC-modified concentration, as shown in Figure 6a. Using the same FIC concentration in the bulk solution and the modified layer can prevent FIC leakage from the modification layer during the test. Figure 6b shows Δ*I* as a function of the glucose concentration for the calculation of the linear range and sensitivity. Moreover, the relationship between Δ*I* and glucose concentration was plotted as Lineweaver–Burk graphs (Figure 6c) to calculate the *K_cat_*/K_m_ ratio values following Equations (3) and (4):(3)$$\frac{1}{\Delta I}=\frac{K_{m}}{\Delta I_{max}}\frac{1}{\left[glucose\right]}\;+\;\frac{1}{\Delta I_{max}}\;$$
(4)$$\Delta I_{max}=k_{cat}{\left[GOx\right]}_{tot}\;$$
where *k_cat_* describes the limiting rate of the enzymatically catalyzed reaction, Δ*I_max_* is the maximal rate of FOC production, and *K_m_* represents the substrate concentration at half of Δ*I_max_*.

Table 2 compares the kinetics parameters of enzymes obtained from the SPECs modified by GOx/FIC/αPLL layers of different concentration ratios. The result shows that the sensitivity obtained at the SPCEs modified with the GOx/FIC/αPLL layer of 0.5/99.5/5 mM is larger than those obtained at the SPCEs modified with the GOx/FIC/αPLL layer of 0.5/99.5/0.5 mM and 0.5/9.5/0.5 mM, implying that increased αPLL concentration improves the electron transfer between GOx and FIC. Moreover, the increasing FIC concentration can significantly improve sensitivity. The higher *K_cat_*/*K_m_* obtained at the SPCEs modified with the GOx/FIC/αPLL layer of 0.5/99.5/5 mM indicates better electron affinity between FIC and GOx/FADH_2_, attributed to entrapment of protonated αPLL with FIC and GOx/FADH_2_.

Furthermore, the synergistic function of αPLL for electron transfer improvement between GOx and FIC was explored by UV–vis spectroscopy. Figure 7 shows the absorption spectra obtained from an αPLL, GOx, FIC, GOx/αPLL mixture and a FIC/αPLL mixture. The absorption peak of αPLL was at about 257 nm. The GOx solution exhibits an absorption peak at 273 nm along with a pair of peaks at 377 nm and 451 nm, representing the FAD and FADH_2_ groups [36]. After αPLL addition, the 273 nm peaks of GOx shifted to 262 nm obtained from the GOx/αPLL mixture, attributed to the interaction of αPLL and GOx. In contrast, the FIC/αPLL mixture had an almost similar absorption spectrum to the FIC solution, suggesting little bonding interaction between αPLL and FIC.

### 3.5. Testing Strips for Real Samples

The optimal GOx/FIC/αPLL ratio of 0.5/99.5/5 mM was used to modify two-electrode SPCEs, and the SPCEs were packaged with a spacer and a hydrophilic cover as blood glucose testing strips by Tyson Bioresearch Inc. A 2 μL aliquot of human serum containing concentration-varied glucose was dripped onto the testing strips and then measured by chronoamperometry. Figure 8 shows the corresponding chronoamperograms. The calibration curve showed good linearity with a linear regression equation of Δ*I* (= *I_glucose_* − *I_blank_*) = 0.212 [glucose] − 0.261 and a correlation coefficient (*R*) of 0.998 in the range of 2.8 mM to 27.5 mM calculated from three individual strips. The statistical background current, *I_blank_*, was 2.92 ± 0.12 µA/mm^2^, implying that the background noise was 0.12 µA/mm^2^. The calculated limit of detection (LOD) was 2.3 mM, based on the IUPAC recommendation (S/N > 3). This linear range can cover the requirement of self-monitoring glucometers for diabetes for use in point-of-care testing [40]. Furthermore, the sensitivity (212.1 nA/mM mm^2^) measured on the testing strips exceeded that (128.7 nA/mM mm^2^) measured at the SPCEs modified with the GOx/FIC/αPLL layer of 0.5/99.5/5 mM in a beaker. This is attributed to the fact that the packaged strips can confine the FOC product in a limiting space to promote the glucose concentration. Table 3 compares the sensing properties of different polymer-based glucose biosensors. The LOD of this study is higher than that obtained from other polymer-based electrochemical glucose biosensors [41,42], attributed to the effect of the small volume (2 μL) of GOx/FIC/αPLL mixture manually dripped onto the two-electrode SPCEs. After packaging, the variation in the amount of modified GOx/FIC/αPLL produces a larger background noise, inducing a larger LOD. If the dripped volume of GOx/FIC/αPLL mixture can be well controlled with an automatic machine, the LOD should be able to be lowered. The sensitivity and linear range obtained in this study are higher and wider than those found in other studies for glucose detection [41,42,43,44,45]. Presently, using metamaterials as a resonator for glucose measurement provides impressive sensing properties, including a wide detecting range of 3 to 30 mM and a low LOD of 0.35 mM [46], but the metamaterial-based biosensor has lower selectivity for glucose detection in a urea-containing solution. That is adverse for the measurement of real blood samples.

### 3.6. Long-Term Stability

The stability of testing strips is one important index in commercialized development. SPCEs coated with 0.5 mM GOx, 99.5 mM FIC, and 0.5 or 5 mM αPLL were packaged as blood glucose testing strips by Tyson Bioresearch Inc. and separately kept at 30 °C and 50 °C for different periods to assess their stability. Keeping strips at 50 °C served to accelerate the aging test. A 2 μL aliquot of 6.67 mM glucose-containing serum was used to estimate the time-lapse change in the oxidative current of FOC. Figure 9 shows the current ratio (I/I_o day_) with the number of storing days. For strips stored at 30 °C, neither the 0.5 mM αPLL- nor the 5 mM αPLL-modified strips exhibited significant I/I_o day_ change over time. Katano also demonstrated that PLL can maintain GOx activity [30]. However, when the strips were stored at 50 °C over 20 days, the 0.5 mM αPLL-modified strips exhibited a significant I/I_o day_ change, but that measured for the 5 mM αPLL-modified strips was still below 10%. The result shows that 5 mM αPLL for the entrapment of GOx and FIC provides better biocompatibility for maintaining GOx activity than does 0.5 mM αPLL.

In this work, the GOx/FIC/αPLL-modified SPCEs packaged as glucose testing strips were demonstrated to successfully detect the glucose concentration of serum samples and exhibit a wide linear range of 2.8–27.5 mM, which covers the ISO15197 regulation of a 2.8–22.2 mM detecting range for a commercial glucometer [47]. Moreover, the good thermal stability of the strips can extend the shelf life of testing strips. The αPLL-based testing strips show great promise for the development of a commercial product to meet the requirement of medical diagnostics of diabetes.

## 4. Conclusions

This study adopted αPLL as the entrapping matrix of GOx and FIC to construct glucose testing strips. The αPLL modification can improve the redox kinetics of FIC and FOC. In particular, tetravalent FOC has a stronger interaction with the protonated αPLL than monovalent Cl^−1^, H_2_PO_4_^−1^, divalent HPO_4_^−2^, and trivalent FIC. The SPCEs coated with 0.5 mM GOx, 99.5 mM FIC, and 5 mM αPLL had greater sensitivity to glucose than did those coated with 0.5 mM GOx, 99.5 mM FIC, and 0.5 mM αPLL, suggesting that the protonated αPLL can promote electron transfer between GOx and FIC. Finally, the SPCEs coated with 0.5 mM GOx, 99.5 mM FIC, and 5 mM αPLL were packaged as blood glucose testing strips to measure glucose-containing human serum samples. The glucose testing strips had good linearity from 2.8 mM to 27.5 mM and a LOD of 2.3 mM. Moreover, the 5 mM αPLL-based glucose testing strips exhibited good long-term stability for the maintenance of GOx activity in a 50 °C aging test. The sensing properties of the glucose strips cover the requirements relating to self-monitoring glucometers for diabetes for use in point-of-care testing. However, the usage of αPLL has only been demonstrated to promote the electrochemical electron transfer efficiency between the mediator, GOx, and SPCEs. More experimental designs are required to explore the effect of the αPLL coating on the optical and electromagnetic properties of SPCEs. Furthermore, the glucose testing strips used in this study need further investigation to realize the effect of varied interferents, such as ascorbic acid, uric acid, acetaminophen, etc., and hematocrit on their sensing properties for the development of commercial products in the future.

## Figures and Tables

**Figure 1 sensors-19-01448-f001:**
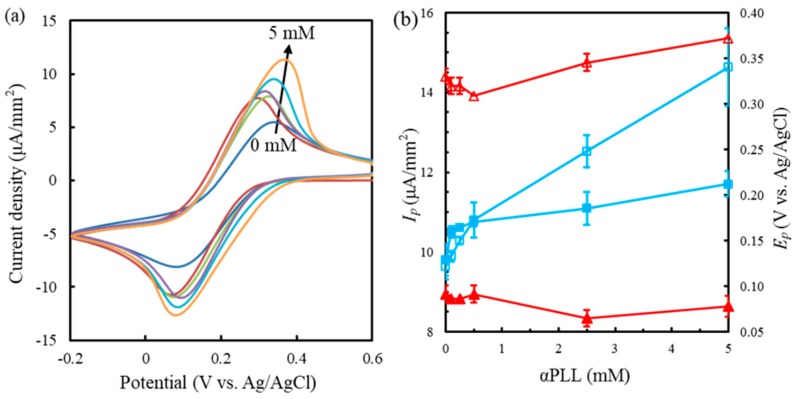
(**a**) Cyclic voltammograms from inner to outer measured at 0, 0.1, 0.25, 0.5, 2.5, and 5 mM α-poly-l-lysine (αPLL)-modified screen-printed carbon electrodes (SPCEs) in 10 mM ferricyanide (FIC) with a scanning rate of 50 mV/s. (**b**) *I_pc_* (solid square), *I_pa_* (empty square), *E_pc_* (solid triangle), and *E_pa_* (empty triangle) as a function of the αPLL concentration coated on the SPCEs (*n* = 3).

**Figure 2 sensors-19-01448-f002:**
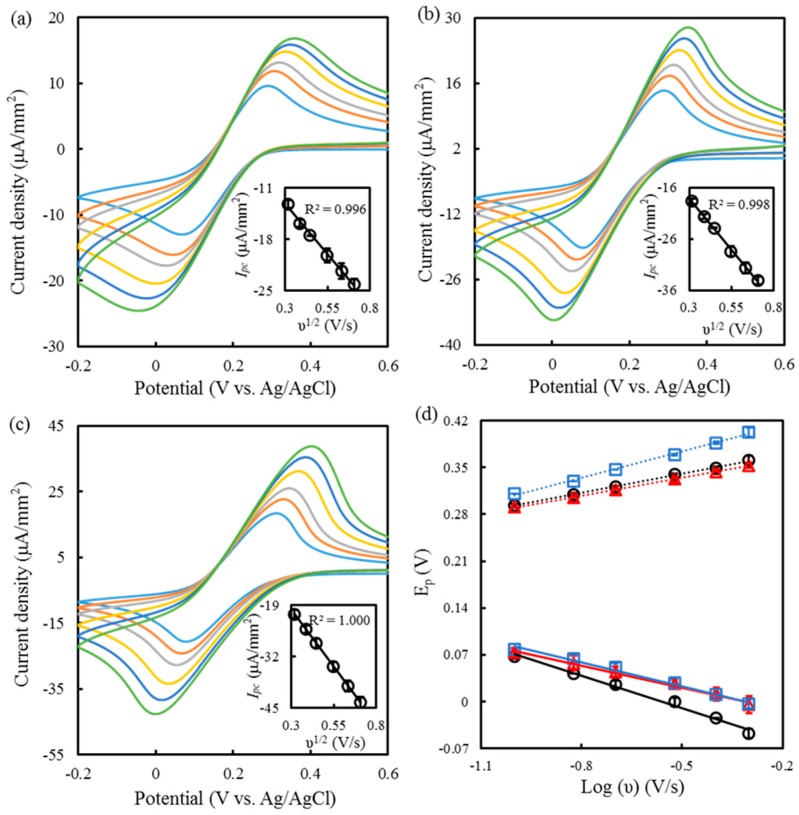
Cyclic voltammograms from inner to outer measured with scanning rates of 100, 150, 200, 300, 400, and 500 mV/s at 0 (**a**), 0.5 (**b**), and 5 mM (**c**) αPLL-modified SPCEs in 10 mM FIC. The insets of (**a**–**c**) show the *I_pc_* value as a function of the square root of the scan rate (*υ*). (**d**) Plots of *E_pc_* (solid line) and *E_pa_* (dash line) versus the logarithm of *υ* at 0 (circle), 0.5 (triangle), and 5 mM (square) αPLL-modified SPCEs.

**Figure 3 sensors-19-01448-f003:**
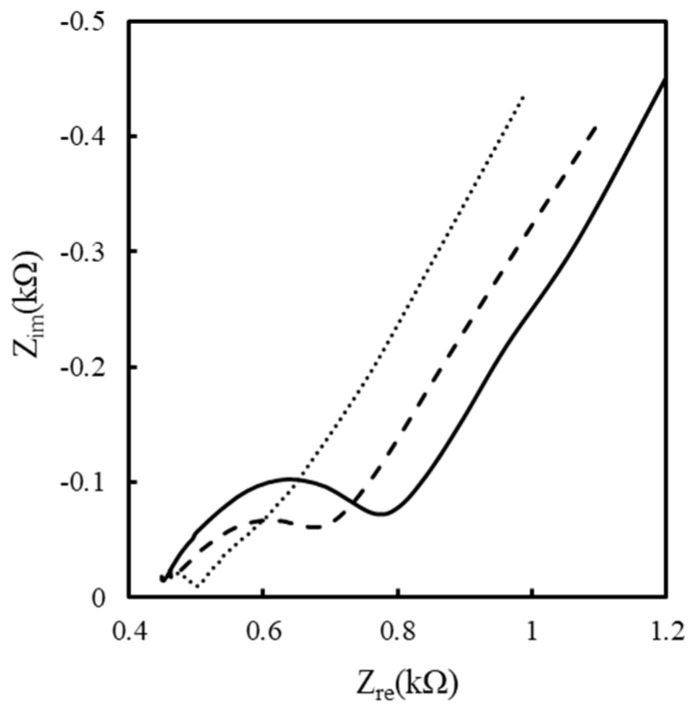
Nyquist plots measured at the 0 (solid line), 0.5 mM (dash line), and 5 mM (dotted line) αPLL-modified SPCEs in a 10 mM FIC/ferrocyanide (FOC) solution.

**Figure 4 sensors-19-01448-f004:**
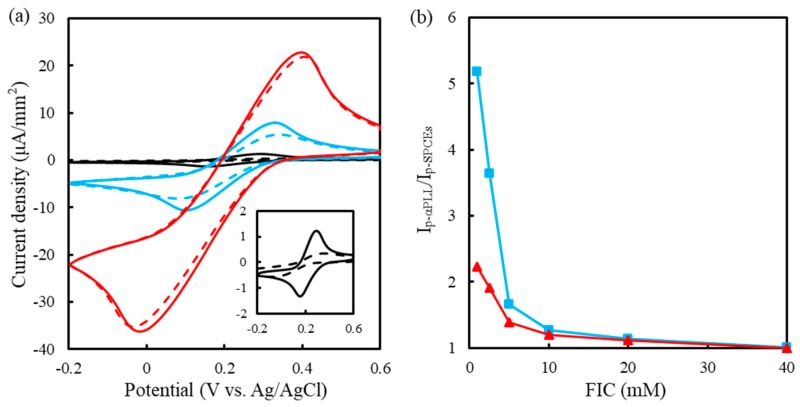
(**a**) Cyclic voltammograms measured at the preoxidized SPCEs without (dash line) and with (solid line) the modification of 0.5 mM αPLL in 1 (black), 10 (blue), and 40 (red) mM FIC solution (pH 7.0) with a scanning rate of 50 mV/s. Insert: 1 mM FIC. (**b**) Plots of *I_pc-PLL_*/*I_pc-SPCE_* (red) and *I_pa-PLL_*/*I_pa-SPCE_* (blue) against FIC concentration (*n* = 3).

**Figure 5 sensors-19-01448-f005:**
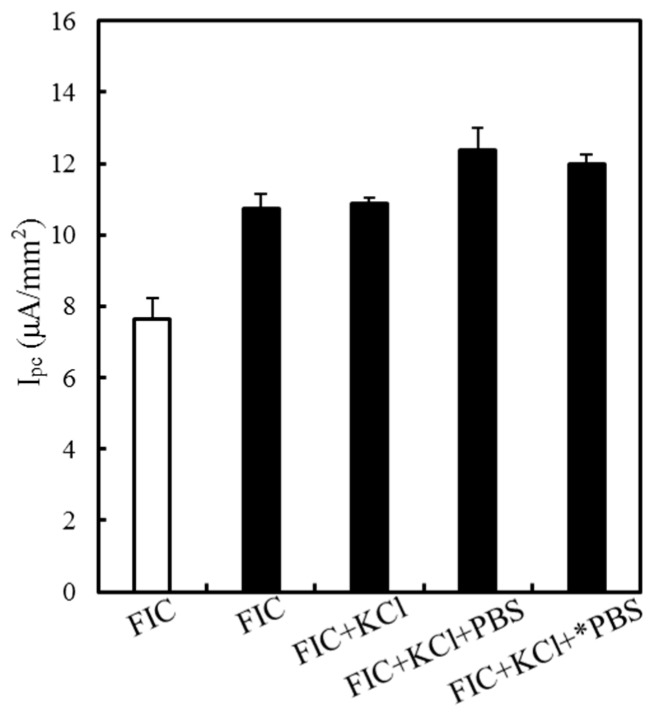
Effect of different anions on the *I_pc_* obtained on the preoxidized SPCEs (white bar) and the 0.5 mM αPLL-modified SPCEs (black bar) in 10 mM FIC (3.79 mS/cm), 10 mM FIC/100 mM KCl (FIC+KCl, 14.39 mS/cm), 10 mM FIC/100 mM KCl/10 mM PBS (FIC+KCl+PBS, 16.76 mS/cm), and 10 mM FIC/100 mM KCl/25 mM PBS (FIC+KCl+*PBS, 19.20 mS/cm) solutions.

**Figure 6 sensors-19-01448-f006:**
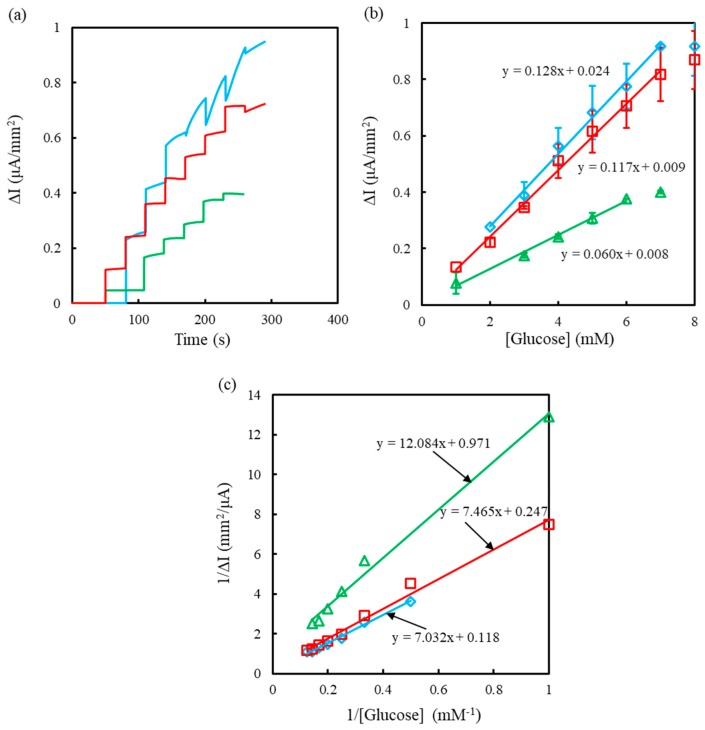
(**a**) Chronoamperograms measured for the SPCEs modified with GOx/FIC/αPLL layers of 0.5/9.5/0.5 mM (green), 0.5/99.5/0.5 mM (red), and 0.5/99.5/5 mM (blue) by successively adding 1 mM glucose into the measuring solution. (**b**) Calibration curves (*n* = 3) obtained for the SPCEs modified with GOx/FIC/αPLL layers of 0.5/9.5/0.5 mM (triangle), 0.5/99.5/0.5 mM (square), and 0.5/99.5/5 mM (rhombus). (**c**) Lineweaver–Burk plots corresponding to (**b**).

**Figure 7 sensors-19-01448-f007:**
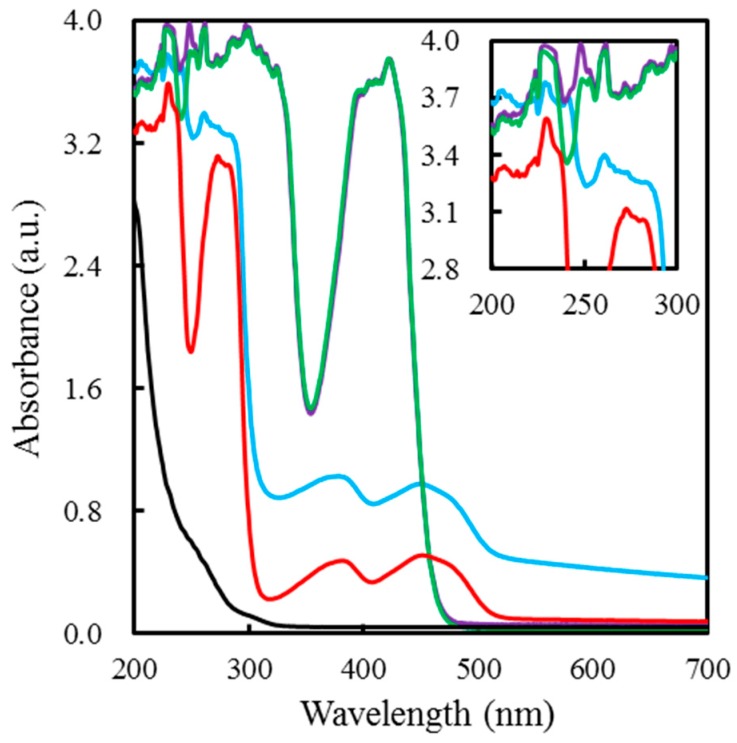
UV−visible absorption spectra of 5 mM αPLL (black), 0.5 mM GOx (red), the mixture of 0.5 mM GOx/5 mM αPLL (blue), 99.5 mM FIC (green), and the mixture of 99.5 mM FIC/5 mM αPLL (purple).

**Figure 8 sensors-19-01448-f008:**
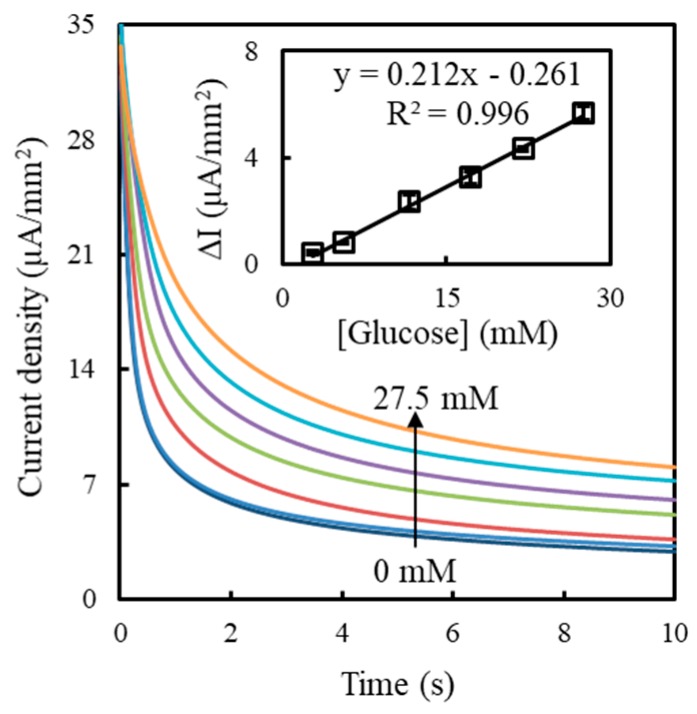
Chronoamperograms obtained for the packaged SPCEs modified with the GOx/FIC/αPLL layer of 0.5/99.5/5 mM as glucose testing strips at an applied potential of +0.25 V in a two-electrode system. Inset shows the Δ*I* (*= I_glucose_* − *I_blank_*) value (*n* = 3) as a function of glucose concentration calculated at 10 s from the chronoamperograms.

**Figure 9 sensors-19-01448-f009:**
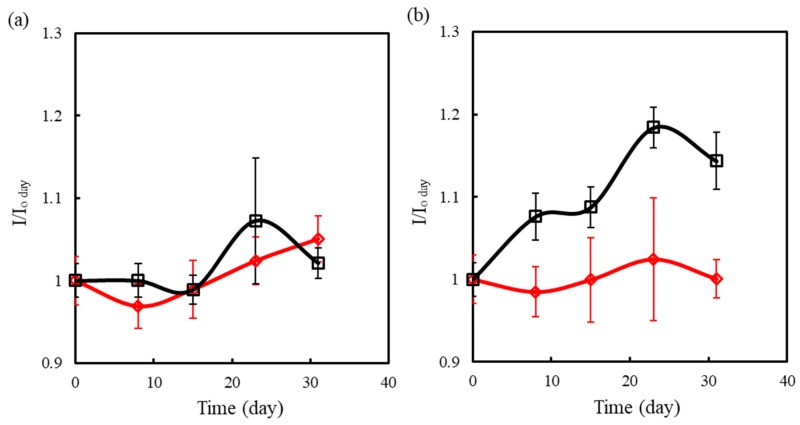
Long-term stability test of the strips modified with a GOx/FIC/αPLL layer of 0.5/99.5/0.5 mM (black line) and 0.5/99.5/5 (red line) when the two types of strips were stored at 30 °C (**a**) and 50 °C (**b**). The current was measured at +0.25 V after dripping 6.67 mM glucose-containing serum on the strips. Each measurement was calculated from three individual strips.

**Table 1 sensors-19-01448-t001:** Values of the circuit elements were fitted using the Randles equivalent circuit for the 0, 0.5, and 5 mM αPLL-modified SPCEs. The experimental parameters are the same as in Figure 3. The statistical values of the mean ± standard deviation were calculated from three repetitions.

SPCEs	*R_et_* (Ω)	*Z_w_* (Ω)	*CPE* (µF)	*α* (×1000)	*R_s_* (Ω)
Preoxidized	318.73 ± 4.47	505.25 ± 1.48	1.34 ± 0.01	659.75 ± 4.32	445.75 ± 1.64
0.5 mM αPLL	237.38 ± 1.98	476.25 ± 3.34	1.57 ± 0.01	570.75 ± 8.20	457.25 ± 0.43
5 mM αPLL	201.93 ± 3.47	875.25 ± 3.27	6.39 ± 0.03	436.75 ± 1.92	486.50 ± 0.50

**Table 2 sensors-19-01448-t002:** Sensing properties and kinetics parameters of enzymes obtained from the different GOx/FIC/αPLL-modified SPECs (*n* = 3). The experimental parameters are the same as those in Figure 6.

Modification of GOx/FIC/αPLL (mM)	Sensitivity (nA/mM mm^2^)	Linear Range (mM)	*k_cat_* (1/s)	*K_m_* (mM)	*k_cat_*/*K_m_*
0.5/9.5/0.5	60.20 ± 5.94	1~6	2.06	12.44	0.166
0.5/99.5/0.5	117.40 ± 20.22	1~7	8.10	30.25	0.268
0.5/99.5/5	128.73 ± 7.46	2~7	16.96	59.64	0.284

**Table 3 sensors-19-01448-t003:** Comparison of limit of detection (LOD), sensitivity, and linear range obtained by different GOx electrodes based on polymer- and derivative-based glucose biosensors.

Electrodes	Sensitivity (nA/mM mm^2^)	Linear Range (mM)	LOD (mM)	Ref.
PAMAM-Fc/GOx/GCEs ^1^	65.4	1.0–22.0	0.48	[41]
Fc–COOH/GOx /Cellulose/SPCEs	-	1.0–5.0	0.18	[42]
Fc/Chi/GOx/GCEs ^2^	8.6	1.0–6.0	-	[43]
PVP-Os/GOx/SPCEs ^3^	120.0	-	-	[44]
GOx/FIC/Pim/MWCNT/GCEs ^4^	0.4	0.3–1.5	-	[45]
GOx/FIC/PLL/SPCEs	212.1	2.8–27.5	2.32	This work

^1^ PAMAM: poly(amido amine). ^2^ Fc/Chi/GOx/GCE: ferrocene/chitosan/GOx/glassy carbon electrode. ^3^ PVP-Os: poly[(vinylpyridine)Os(bipyridyl)_2_Cl^2+/3+^]. ^4^ GOx/FIC/Pim/MWCNT/GCE: GOx/FIC/imidazolium-based polymer/multiwalled carbon nanotubes/GCEs.

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
