# Peer review of "Effect of Poly-l-Lysine Polycation on the Glucose Oxidase/Ferricyanide Composite-Based Second-Generation Blood Glucose Sensors"

_sensors, 2019, doi:10.3390/s19061448_

Reviewer 1 Report

The paper is clear and pretty well written. The topic is very interesting.
On the other hand, some parts are too much descriptive and very few rigorous.
In the following, my comments to address before acceptance:

_ Section "1. Introduction"

The introduction is really clear and all the objectives well stated.
On the other hand some recent technology developments are missing such as:

_ nanoparticles [Metamaterial resonator arrays for organic and inorganic compound sensing, Photonics, Devices, and Systems V 8306, 83060I, 2011]
_ near-zero-index materials [Near-zero refractive index photonics, Nature Photonics, 11, 149–158, 2017]
_ meta-surfaces [Metasurfaces for Advanced Sensing and Diagnostics, Sensors 2019, 19(2), 355, 2019]
_ graphene [Transformation optics using graphene, Science, 332(6035),  1291-1294, 2011]
_ plasmons [Spectral Green's function for SPR meta-structures, Materials Science Forum 792, 2014]

It would be beneficial if authors can expand and explain such techniques and make a comparison among theirs and the above suggested works: pro/cons should be highlighted.

_ Section "3. Results and Discussion"

The related method is very well detailed. On the other hand, to explore the structure behavior:

1) Authors should refer to the following additional phenomena such as: electric currents, surface waves and magnetic currents.

2) The paper lack in further application examples. Besides the glucose application. Take into consideration the following:medical diagnostics, telemedicine, refractive index measurements, nanotherapy and antennas.
I would suggest to create a small paragraph by considering such applications and explaining how you can use your device for them.
Please highlight what's new in yours.

_ Section "4 Conclusions"

1) No limitations of the proposed method have been highlighted.

2) No future improvements/works have been discussed.

Author Response

We really appreciate that you can give us an opportunity to revise the manuscript. Thank you very much for the comments and suggestions to improve the quality of this manuscript. We provide a point-by-point response to the reviewer’s comments below. The revised descriptions were marked by red color.

--------------------------------------------------------------------------------------------------------------------

Response to Reviewer 1 Comments

The paper is clear and pretty well written. The topic is very interesting.

On the other hand, some parts are too much descriptive and very few rigorous.

In the following, my comments to address before acceptance:

_ Section "1. Introduction"

The introduction is really clear and all the objectives well stated.

On the other hand some recent technology developments are missing such as:

_ nanoparticles [Metamaterial resonator arrays for organic and inorganic compound sensing, Photonics, Devices, and Systems V 8306, 83060I, 2011]

_ near-zero-index materials [Near-zero refractive index photonics, Nature Photonics, 11, 149–158, 2017]

_ meta-surfaces [Metasurfaces for Advanced Sensing and Diagnostics, Sensors 2019, 19(2), 355, 2019]

_ graphene [Transformation optics using graphene, Science, 332(6035),  1291-1294, 2011]

_ plasmons [Spectral Green's function for SPR meta-structures, Materials Science Forum 792, 2014]

It would be beneficial if authors can expand and explain such techniques and make a comparison among theirs and the above suggested works: pro/cons should be highlighted.

Ans: Thanks for your suggestion. We have added the literatures you suggested in Introduction on page 1 lines 32-36, “Presently, several methods, including metamaterial-based electromagnetic spectroscopy [4], fluorescence [5], near infrared spectroscopy [6], and electrochemistry [7], have been developed for glucose detection. Furthermore, most commercial glucose biosensors adapt electrochemical methods via the catalysis of glucose oxidase (GOx) or glucose dehydrogenase to specifically detect glucose concentration [8].”, and compared the sensing properties of metamaterial-based glucose sensor with this study in Results and Discussion on page 10 lines 328-331, “Presently, using metamaterials as an resonator for glucose measurement has impressive sensing properties, including the wide detecting range of 3 to 30 mM and the low LOD of 0.35 mM [46], but the metamaterial-based biosensor had less  selectivity for glucose detection in a urea-containing solution. That is adverse for the measurement of real blood samples.” .

_ Section "3. Results and Discussion"

The related method is very well detailed. On the other hand, to explore the structure behavior:

1) Authors should refer to the following additional phenomena such as: electric currents, surface waves and magnetic currents.

Ans: Thanks for your suggestion. This study focuses on the αPLL interaction with GOx and FIC to promote the electric current response of second-generation glucose biosensors. Therefore, the electrochemical properties of sensor, including peak current, peak potential, electron-transfer kinetics and enzymatic kinetics, have been discussed in this manuscript in detail. Using optical and electromagnetic methods to detect the surface waves and magnetic current of screen-printed carbon electrodes is a good suggestion, we will try to explore these issues in next work. Furthermore, in order to explore the interaction between αPLL, GOx and FIC, UV-visible spectroscopy has been used and the corresponding method and results are added on page 3, “2.4. UV-visible spectroscopy

Ultraviolet-visible (UV−visible) spectrum was recorded with a microplate spectrophotometer (Epoch™, Biotek, USA). 5 µL aliquot of GOx, αPLL, FIC, GOx/αPLL mixture and FIC/αPLL mixture was respectively dripped on a micro-volume plate to determine the absorption spectrum.”, and on page 9, “Furthermore, the synergistic function of αPLL for the electron-transfer improvement between GOx and FIC was explored by UV-vis spectroscopy. Figure 7 shows the absorption spectra obtained from αPLL, GOx, FIC, GOx/αPLL mixture and FIC/αPLL mixture. The absorption peaks of αPLL was about 257 nm. The GOx solution exhibits an absorption peak at 273 nm along with a pair of peaks at 377 nm and 451 nm, representing FAD and FADH2 groups [36]. After αPLL addition, the 273 nm peaks of GOx shifted to 262 nm obtained from the GOx/αPLL mixture, attributed to the interaction of αPLL and GOx. In contrast, the FIC/αPLL mixture had almost similar absorption spectrum with the FIC solution, suggesting little bonding interaction between αPLL and FIC.

Figure 7. UV−visible absorption spectra of 5 mM αPLL (black), 0.5 mM GOx (red), the mixture of 0.5 mM GOx/5 mM αPLL (blue), 99.5 mM FIC (green) and the mixture of 99.5 mM FIC/5 mM αPLL (purple).

2) The paper lack in further application examples. Besides the glucose application. Take into consideration the following: medical diagnostics, telemedicine, refractive index measurements, nanotherapy and antennas.

I would suggest to create a small paragraph by considering such applications and explaining how you can use your device for them.

Please highlight what's new in yours.

Ans: Thank you for the suggestion. Second-generation glucose biosensors are the mainstream of commercial glucometers. Moreover, the screen-printed carbon electrodes are the most used testing strips. The GOx/FIC/αPLL-based strips of this study can be used for quantifying the blood glucose concentration of patients in medical diagnostics. This suggestion has been mentioned on page 11 lines 357-362, “In this work, the GOx/FIC/αPLL-modified SPCEs packaged as glucose testing strips were demonstrated to successfully detect the glucose concentration of serum samples and exhibit a wide linear range of 2.8-27.5 mM, which has covered the ISO15197 regulation with 2.8-22.2 mM detecting range for a commercial glucometer [47]. Moreover, the good thermal stability of the strips can extend the shelf life of testing strips. The αPLL-based testing strips have great promise to develop a commercial product for the requirement of medical diagnostics of diabetes.”.

 _ Section "4 Conclusions"

1) No limitations of the proposed method have been highlighted.

2) No future improvements/works have been discussed.

Ans: Thanks for your suggestion. The limitation and future improvement of this work is added on page 12 lines 382-388, “However, the usage of αPLL is only demonstrated to promote the electrochemical electron-transfer efficiency between mediator, GOx and SPCEs. More experimental designs are required for exploring the effect of αPLL coating on the optical and electromagnetic properties of SPCEs. Furthermore, the glucose testing strips of the study needs further estimation to realize the effect of varied interferents, such as ascorbic acid, uric acid, acetaminophen etc., and hematocrit on the sensing properties for the development of commercial product in the future.

Reviewer 2 Report

This is a paper about the effect of poly-L-lysine on glucose oxidase/ferricyanide composite-based blood glucose sensors. The paper is written in good English, well structured, and findings are underlined with systematic electrochemical experiments. T

After having read the paper I have two major concerns that have to be addressed before publication can be recommended:

First, the novelty of the paper is not clear to me. Katano and his group published several papers in this field. The latest in 2014 and 2018 in Analytical Sciences were not cited. (Please insert a citation). Please make clear what the novelty of your paper is as compared to that published by other groups.

Second: For systematic studies of the effect of poly-L-lysine on glucose oxidase/ferricyanide composite-based blood glucose sensors in state-of-art testing strips, you should make sure that your number of experiments is large enough to give a scientifically sound picture. A sample number of n = 3 is not sufficient. 

3.4

·        what is your n?

 3.5

·        Fig. 7 the number of examples used for the experiment is only n=3. 

3.6 

·        Fig. 8 the number of examples used for the experiment is only n=3. 

·        l.325: how do you link biocompatibility and temperature stability.

Summary conclusion

·        please discuss the fact that your LOD is very high as compared to  other work (Tab. 3)

Author Response

We really appreciate that you can give us an opportunity to revise the manuscript. Thank you very much for the comments and suggestions to improve the quality of this manuscript. We provide a point-by-point response to the reviewer’s comments below. The revised descriptions were marked by red color.

----------------------------------------------------------------------------------------------------------------------------------

Response to Reviewer 2 Comments

This is a paper about the effect of poly-L-lysine on glucose oxidase/ferricyanide composite-based blood glucose sensors. The paper is written in good English, well structured, and findings are underlined with systematic electrochemical experiments.

After having read the paper I have two major concerns that have to be addressed before publication can be recommended:

First, the novelty of the paper is not clear to me. Katano and his group published several papers in this field. The latest in 2014 and 2018 in Analytical Sciences were not cited. (Please insert a citation). Please make clear what the novelty of your paper is as compared to that published by other groups.

Ans: Thanks for your suggestion. The studies published by Katano et al. are cited in the refs. 13 (in 2014), 16 (in 2012), 28 (in 2008), 29 (in 2009), 30 (in 2011), 31 (in 2018) and 37 (in 2008). The difference between this study and katano’s studies is compared as below.

Katano et al. (2014) used a glassy carbon electrode in the solution of εPLL, GOD and ferrocene derivatives mixture to explore the effect of suspended εPLL on the electron transfer kinetics between GOx and ferrocene derivatives having different ionic charges, as mentioned on page 1 line 40. They only discussed the electrostatic interaction between εPLL, GOD and negatively charged ferrocene derivatives, and didn’t develop a εPLL-modified glucose sensor and explore the sensing properties of sensors.

On page 2 lines 60-64 these sentences have mentioned the 2018-year findings of Uematsu and Katano [31] and the limitation in practical blood measurement. “Uematsu et al. found that a glucose biosensor fabricated by covalently entrapping GOx in a εPLL matrix via glutaraldehyde on a glassy carbon electrode had an improved catalytic response in a more acidic solution, such as the pH 5.0 acetate buffer [31]. Although Uematsu et al. proved that the sensitivity of εPLL/GOx-modified sensor was larger than that of αPLL/GOx-modified sensor, the current response measured at the εPLL/GOx-modified sensor was significantly affected by varied pH values [31]. Compared with εPLL (pKa = 7.6) [16], αPLL (pKa = 10.3) possesses more protonation and higher solubility in a physiological solution (pH 7.4).” Moreover, due to good biocompatibility, high solubility and easy protonation in a physiological solution, αPLL has been used as an entrapping matrix for glucose biosensors [32, 33].”.

Therefore, in this study αPLL was used as the entrapment matrix for the immobilization of negatively charged GOx and FIC on screen printed carbon electrodes (SPCEs) to construct disposable glucose biosensors and examined the effect of αPLL on the sensing properties of GOx/FIC composite-based second generation glucose biosensor. In the Uematsu study (2018) they didn’t combine the εPLL PLL/GOx modification with a disposable electrode to form a sensing strip. In contrast, the purpose of this study matches the topic, “smart electrochemical screen-printed platforms”, of Sensors special issue.

Second: For systematic studies of the effect of poly-L-lysine on glucose oxidase/ferricyanide composite-based blood glucose sensors in state-of-art testing strips, you should make sure that your number of experiments is large enough to give a scientifically sound picture. A sample number of n = 3 is not sufficient.

3.5   What is your n?

3.6   Fig. 8 the number of examples used for the experiment is only n=3.

3.7   Fig. 9 the number of examples used for the experiment is only n=3.

Ans: In this study all data was statistically obtained from at least three separate electrodes, and mentioned in Figure caption. In the experiments, we prepared 28 independent packaged SPCEs for the calibration curve of Fig. 8 and 80 independent packaged SPCEs for the stability of Fig. 9, and collected the better three SPCEs for each testing point to obtain the statistical values. The three repetitions (n =3) for each testing point is often reported in different papers published in Sensors. For example, Rassas et al., (2019) showed the calibration curve and the reproducibility with three independent electrodes. The statistic value can be convinced.

Reference

Rassas, I., Braiek, M., Bonhomme, A., Bessueille, F., Raffin, G., Majdoub, H., Jaffrezic-Renault, N., Highly Sensitive Voltammetric Glucose Biosensor Based on Glucose Oxidase Encapsulated in a Chitosan/Kappa-Carrageenan/Gold Nanoparticle Bionanocomposite. Sensors 2019, 19.

l.354: how do you link biocompatibility and temperature stability?

Ans: The definition of biocompatibility is the ability of a material to perform with an appropriate host response in a specific situation (https://en.wikipedia.org/wiki/Biocompatibility). Thermal treatment may induce the protein denaturation. If the addition of αPLL can reduce GOx denaturation to maintain the same enzymatic activity, the αPLL material shows good biocompatibility for GOx.

Summary conclusion

Please discuss the fact that your LOD is very high as compared to other work (Tab. 3)

Ans: thanks for your suggestion. We have explained the result on page 10 lines 321-326. “Table 3 compares the sensing properties of different polymer-based glucose biosensors. “The LOD of this study is higher than that obtained from other polymer-based electrochemical glucose biosensors [41, 42], attributed to the effect of small volume (2 mL) of GOx/FIC/αPLL mixture manually dripped on the two-electrode SPCEs. After package, the variation in the amount of modified GOx/FIC/αPLL produces a larger background noise, inducing a larger LOD. If the dripped volume of GOx/FIC/αPLL mixture can be well controlled with an automatic machine, the LOD should be able to be lowered             

Round  2

Reviewer 2 Report

The authors improved the script according to my comments.